

# The prognostic role of lymphocyte-to-monocyte ratio in patients with resectable pancreatic cancer: a systematic review and meta-analysis

Haipeng Li[1], Shang Peng[2], Ran An[3], Nana Du[2], Huan Wu[1], Xiangcheng Zhen[4], Yuanzhi Gao[4], Zhenghong Li[3] and Jingting Min[2]

[1] Department of Mental Health, Bengbu Medical College, Bengbu, Anhui, China
[2] Department of Basic Medicine, Bengbu Medical College, Bengbu, Anhui, China
[3] Department of Life Science, Bengbu Medical College, Bengbu, Anhui, China
[4] Department of Clinical Medicine, Bengbu Medical College, Bengbu, Anhui, China

## ABSTRACT

**Objectives:** This systematic review and meta-analysis examined whether the lymphocyte-to-monocyte ratio (LMR) can serve as an indicator for predicting the prognosis of patients with resectable pancreatic cancer.

**Patients and Methods:** This meta-analysis was registered with PROSPERO: CRD42023461260. A systematic literature search was conducted in the PubMed, Embase, Cochrane, and Web of Science databases up to September 2023 to assess whether LMR can predict the prognosis of patients with resectable pancreatic cancer. The outcomes measured included subgroup analyses of overall survival (OS) with hazard ratios (HR) and confidence intervals of geographical region, patient population, and LMR threshold. A sensitivity analysis was also performed for OS and HR and confidence intervals were calculated for recurrence-free survival (RFS).

**Results:** A total of 14 eligible articles, comprising 4,019 patients, were included in the comprehensive analysis. The results of this comprehensive analysis indicate that LMR is a robust predictor of OS, demonstrating strong prognostic significance (HR = 0.55, 95% CI [0.44–0.69], $I^2$ = 79%, $P < 0.00001$). This predictive significance extended to various types of pancreatic cancer, such as pancreatic ductal adenocarcinoma (HR = 0.73, 95% CI [0.57–0.93], $I^2$ = 46%, $P = 0.01$), pancreatic neuroendocrine neoplasms (HR = 0.81, 95% CI [0.66–0.99], $P = 0.04$) and other subtypes (HR = 0.40, 95% CI [0.22–0.72], $I^2$ = 89%, $P < 0.00001$), but not to pancreatic head cancer (HR = 0.46, 95% CI [0.16–1.13], $I^2$ = 59%, $P = 0.12$). LMR retained its predictive value across different regions, including Asia (HR = 0.62, 95% CI [0.47–0.76], $I^2$ = 68%, $P < 0.0001$), Europe (HR = 0.78, 95% CI [0.67–0.91], $I^2$ = 0%, $P = 0.002$), and the Americas (HR = 0.14, 95% CI [0.08–0.24], $I^2$ = 0%, $P < 0.00001$). Notably, both LMR cut-off values greater than or equal to three (HR = 0.62, 95% CI [0.47–0.82], $I^2$ = 67%, $P = 0.0009$) and less than three (HR = 0.47, 95% CI [0.32–0.69], $I^2$ = 85%, $P = 0.0001$) exhibited prognostic significance. The sensitivity analysis for OS confirmed the strong predictive value of LMR, whereas LMR did not exhibit predictive significance for RFS (HR = 0.35, 95% CI [0.09–1.32], $I^2$ = 95%, $P = 0.12$). In both subgroups categorized by Newcastle-Ottawa Scale (NOS) scores of ≥7 (HR = 0.66, 95% CI [0.54–0.80], $I^2$ = 53%, $P = 0.04$) and <7

Corresponding authors
Zhenghong Li, lzhbbmc@126.com
Jingting Min, 429118736@qq.com

(HR = 0.41, CI [0.23–0.72], $I^2$ = 89%, $P$ < 0.00001), LMR was demonstrated to have predictive value.

**Conclusion:** Despite the observed heterogeneity and potential biases in the included studies, the findings of this study suggest that LMR may serve as a valuable predictor of OS in patients with resectable pancreatic cancer.

# INTRODUCTION

Pancreatic cancer is one of the most aggressive forms of malignancy and is characterized by a grim prognosis. Only a small subset of pancreatic cancer patients, ranging from 10% to 20%, meet the criteria for surgical intervention at the time of their initial diagnosis. Surgical resection stands as the sole curative approach in pancreatic cancer, providing a modest 5-year survival rate of approximately 30% (*Strobel et al., 2019*).

*Hanahan & Weinberg (2011)* reorted on the gradual acquisition of distinct traits by cancer cells, a process that persists across all stages of cancer evolution, including progression and metastasis. Among these traits is the promotion of inflammation by the cancer itself. These hallmarks of cancer serve as the foundation for contemporary global cancer research (*Balkwill & Mantovani, 2001*; *Colotta et al., 2009*; *Diakos et al., 2014*). Various studies have suggested that serum inflammatory biomarkers, including neutrophils (*Uribe-Querol & Rosales, 2015*), lymphocytes (*Kitayama et al., 2010*), monocytes (*Zhang et al., 2015*), platelets (*Menter et al., 2014*), and albumin (*Gupta & Lis, 2010*), hold the potential to predict prognosis across various cancer types.

In individuals diagnosed with various malignancies, a multitude of biomarkers linked to the inflammatory response have demonstrated promising prognostic capabilities. These factors encompass the neutrophil-to-lymphocyte ratio, the platelet-to-lymphocyte ratio, and the lymphocyte-to-monocyte ratio (LMR) (*Jamieson et al., 2011*; *Shirai et al., 2015*; *An et al., 2010*). *Fujiwara et al. (2014)* performed an analysis that encompassed 111 patients diagnosed with pancreatic cancer who had undergone pancreatic resection. Post-surgical patients with a lower peripheral LMR exhibited significantly worse prognostic outcomes and overall survival rates than post-surgical patients who had a higher peripheral LMR (*Fujiwara et al., 2014*). In a recent study, *Riauka, Ignatavicius & Barauskas (2020)* reported on the prognostic significance of the preoperative platelet-to-lymphocyte ratio (PLR) in resectable pancreatic cancer. They concluded that PLR is a predictive factor for disease-free survival (DFS) in patients with resectable pancreatic cancer (*Riauka, Ignatavicius & Barauskas, 2020*).

However, the existing body of literature investigating the correlation between preoperative LMR and the survival of patients undergoing resection for pancreatic adenocarcinoma is relatively limited. This study endeavors to provide insights into the predictive capacity of LMR in assessing the prognosis of patients with resectable pancreatic cancer by scrutinizing LMR ratios.

## METHODS

### Literature search

This systematic analysis adhered to the Preferred Reporting Items for Systematic Reviews and Meta-Analyses (PRISMA) 2020 guidelines (*Page et al., 2021*) and relied on data registered in PROSPERO under the reference CRD42023461260. The PRISMA 2020 checklist can be found within Table S1.

In September 2023, a systematic literature search of English-language publications was conducted using PubMed, Embase, Cochrane, and Web of Science to investigate the prognostic role of the lymphocyte-to-monocyte ratio in resectable pancreatic cancer. Detailed search methods are outlined in Table S2.

Additionally, a manual review of the reference lists of all eligible studies was performed. Two investigators independently conducted searches for nested reports within primary studies (Haipeng Li and Shang Peng). If contrasting opinions arose, the two investigators discussed until a consensus was reached. In cases where a consensus was not reached, a third investigator (Ran An) was included to facilitate resolution.

### Identification of eligible studies

The final analysis included studies that fulfilled the following criteria: (1) were randomized controlled trials, cohort, or case-control studies; (2) involved adult patients with pancreatic cancer; (3) reported LMR; (4) involved patients eligible for surgical removal; and (5) had sufficient data available to calculate HR.

The following types of publications were excluded: reviews, letters, editorial comments, case reports, conference abstracts, pediatric articles, unpublished articles, and articles not in English. Additionally, patients with advanced pancreatic cancer were excluded as surgical removal is not feasible for this patient population. For studies that provided LMR data, a reciprocal transformation method was applied, as necessary.

### Data extraction

Data extraction was independently conducted by two investigators. In cases of discordance, a third investigator was consulted to make a final determination. The following data was extracted from the studies included in the analysis: primary author, publication year, study duration, study location, sample size, participant age, TNM stage, LMR threshold, body mass index (BMI), NOS, and patient eligibility for surgical removal. In instances where the study reported continuous variables as a median with a range or interquartile range, a validated mathematical approach was applied to compute the mean ± standard deviation (*Wan et al., 2014*; *Luo et al., 2018*). For the studies with absent or unreported data, the investigators proactively reached out to the corresponding authors for the provision of complete data if it was accessible. Detailed information can be found in Table 1.

### Quality assessment

The quality of the incorporated studies was evaluated using the Newcastle-Ottawa Scale (NOS) (*Gurol-Urganci et al., 2013*), with studies receiving seven to nine points considered

**Table 1 Characteristics of eligible studies and assessment of risk of bias.**

| Authors | Journal | Study period | Region | Study design | Population | No. of patients | Gender Male | Female | Age | BMI | TNM stage | LMR threshold | OS | RFS | NOS | Eligible for surgical removal |
|---|---|---|---|---|---|---|---|---|---|---|---|---|---|---|---|---|
| Neumann et al. (2023) | Cancers | 2009–2021 | Germany | Retrospective cohort | Pancreatic cancer | 1,294 | 718 | 576 | 66 (28–94) | NA | NA | 1.6 | 0.8 (0.7–0.99) | NA | 6 | Receive surgical treatment |
| Kubota et al. (2022) | Ann Surg Oncol | 2004–2020 | Japan | Retrospective cohort | Pancreatic ductal adenocarcinoma | 170 | 94 | 76 | NA | 69 (35–88) | NA | 3.3 | 0.284 (0.154–0.524) | NA | 7 | Receive surgical treatment |
| Ueberroth et al. (2021) | J Gastrointest Cancer | NA | America | Retrospective cohort | Pancreatic cancer | 27 | NA | NA | 59.3 ± 10.2 | 27.0 ± 6.2 | NA | 2.5 | 0.743 (0.429–1.289) | NA | 6 | Receive surgical treatment |
| Markus et al. (2021) | J Cancer Res Clin Oncol | 2005–2019 | Germany | Retrospective cohort | Pancreatic ductal adenocarcinoma | 193 | 103 | 90 | 63 (31–89) | NA | I–IV | 2.8 | 0.538 (0.347–0.834) | NA | 6 | Receive surgical treatment |
| Fang et al. (2021) | Cancer Res Treat | 2008–2014 | China | Retrospective cohort | locally and metastatic PC patients | 534 | 317 | 217 | NA | NA | NA | 2.8 | 0.801 (0.653–0.983) | NA | 7 | Receive surgical treatment |
| Zhou et al. (2020) | Endocrine Connections | 2008–2018 | China | Retrospective cohort | Pancreatic neuroendocrine neoplasms | 174 | 82 | 92 | 53 (43–61) | NA | I–IV | 5 | 0.57 (0.13–2.38) | 0.30 (0.11–0.85) | 7 | Receive surgical treatment |
| Takeuchi et al. (2020) | Anticancer Res | 2006–2016 | Japan | Retrospective cohort | Pancreatic head cance | 32 | 17 | 15 | 75 | NA | III–IV | 3 | 0.21 (0.06–0.67) | NA | 7 | Receive surgical treatment |
| Takano et al. (2020) | BMC Cancer | 2006–2016 | Japan | Retrospective cohort | PDAC, PACC and IPMN | 28 | 17 | 11 | 67.5 ± 12 | NA | NA | 3.3 | 0.07 (0.01–0.37) | NA | 7 | Receive surgical treatment |
| Pointer et al. (2020) | BMC Cancer | 2007–2015 | America | Retrospective cohort | Pancreatic ductal adenocarcinoma | 277 | 157 | 120 | 68.0 | 26.3(16.7–58.5) | NA | 2.9 | 0.82 (0.63–1.08) | 0.90 (0.69–1.16) | 7 | Receive surgical treatment |
| Onoe et al. (2019) | Med Princ Pract | 2005–2016 | Japan | Retrospective cohort | Pancreatic head cancer | 165 | 89 | 76 | 72 (45–83) | 20.8 (16.4–30.2) | I–IV | 2.8 | 0.581 (0.346–0.98) | NA | 6 | Receive surgical treatment |
| Kawai et al. (2019) | Surgery | 2010–2016 | Japan | Retrospective cohort | Borderline resectable pancreatic cancer | 65 | 38 | 27 | 70 | NA | I–III | 3 | 0.402 (0.216–0.746) | NA | 6 | Receive surgical treatment |
| Sierzega et al. (2017) | Ann Surg Oncol | 1990–2012 | Poland | Retrospective cohort | Pancreatic ductal adenocarcinoma | 442 | 260 | 182 | 60 (55–66) | NA | I–III | 3 | 0.606 (0.388–0.943) | NA | 5 | All tumor-free margins were at least 1 mmwide |
| Li et al. (2016) | Onco Targets Ther | 2012–2014 | China | Retrospective cohort | Primary pancreatic adenocarcinoma | 144 | 82 | 62 | 60 | NA | I–III | 2.8 | 0.148 (0.085–0.252) | 0.152 (0.092–0.250) | 8 | Receive surgical treatment |
| Stotz et al. (2015) | Clin Chem Lab Med | 2004–2012 | Austria | Retrospective cohort | Pancreatic ductal adenocarcinoma | 474 | 256 | 218 | 65 | NA | I–IV | 2.8 | 0.81 (0.66–0.99) | NA | 6 | Receive surgical treatment |

**Note:**
PDAC, Pancreatic ductal adenocarcinoma ; PACC, pancreatic acinar cell carcinoma; IPMN, intraductal papillary mucinous neoplasm

to be high-quality studies and four to six points considered to be moderate-quality studies (*Kim et al., 2019*). Additionally, the level of evidence for each study was assessed following the guidelines established by the Oxford Centre for Evidence-Based Medicine Levels of Evidence Working Group. Two investigators independently appraised the quality of each eligible study and assigned it an evidence level. Any disparities were resolved through discussion between the two investigators. Detailed NOS results can be found in Table S3.

## Statistical analysis

An evidence synthesis was performed using Review Manager version 5.3, developed by the Cochrane Collaboration in Oxford, UK. Hazard ratio (HR) was used for comparing continuous and dichotomous variables. All metrics were presented with corresponding 95% confidence intervals (CIs). The chi-squared test (Cochran's Q) and the inconsistency index ($I^2$) were used to assess the heterogeneity of the included studies (*Higgins & Thompson, 2002*). Significant heterogeneity was indicated by an $\chi$ $p$-value < 0.05 or $I^2$ > 50%. In cases where significant heterogeneity was identified, a random-effects model was applied for the estimation of the combined HR; otherwise, a fixed-effect model was applied. One-way sensitivity analyses were also performed to gauge the impact of individual studies on the combined results for outcomes characterized by significant heterogeneity. Additionally, for outcomes that encompassed ten or more included studies, Egger's regression tests were performed using Stata version 15.0 (Stata Corp, College Station, TX, USA). A $p$-value < 0.05 was considered statistically significant publication bias (*Egger et al., 1997*).

## RESULTS

### Literature search and study characteristics

The systematic search and selection process is presented as a flowchart in Fig. 1. The systematic literature search yielded a total of 355 relevant articles, with contributions from various sources including PubMed ($n$ = 103), Embase ($n$ = 143), Cochrane ($n$ = 3), and Web of Science ($n$ = 106). After duplicate publications were removed, a total of 224 titles and abstracts were reviewed for inclusion. Ultimately, 14 full-text articles, encompassing data from 4,019 patients, were included in the final pooled analysis and among them, the study by Ueberroth et al. adjusted for sex, age, and stage at diagnosis, while the remaining studies did not make adjustments (*Stotz et al., 2015*; *Li et al., 2016*; *Sierzega et al., 2017*; *Kawai et al., 2019*; *Onoe et al., 2019*; *Pointer et al., 2020*; *Takano et al., 2020*; *Takeuchi et al., 2020*; *Zhou et al., 2020*; *Fang et al., 2021*; *Markus et al., 2021*; *Ueberroth et al., 2021*; *Kubota et al., 2022*; *Neumann et al., 2023*). It is noteworthy that all 14 included studies used a retrospective cohort study design. Of the final 14 articles, eight originated from Asia, four from Europe, and two from the Americas. LMR values ranged between 1.6–3.3, most of the thresholds were around three, and the NOS scores ranged between 6–8 points. LMR was a robust predictor of OS, demonstrating strong prognostic significance (HR = 0.55, 95% CI [0.44–0.69], $I^2$ = 79%, $P$ < 0.00001). This predictive significance extended to various types of pancreatic cancer, such as pancreatic ductal adenocarcinoma (HR = 0.73, 95% CI [0.57–0.93], $I^2$ = 46%, $P$ = 0.01), pancreatic neuroendocrine neoplasms (HR = 0.81, 95% CI
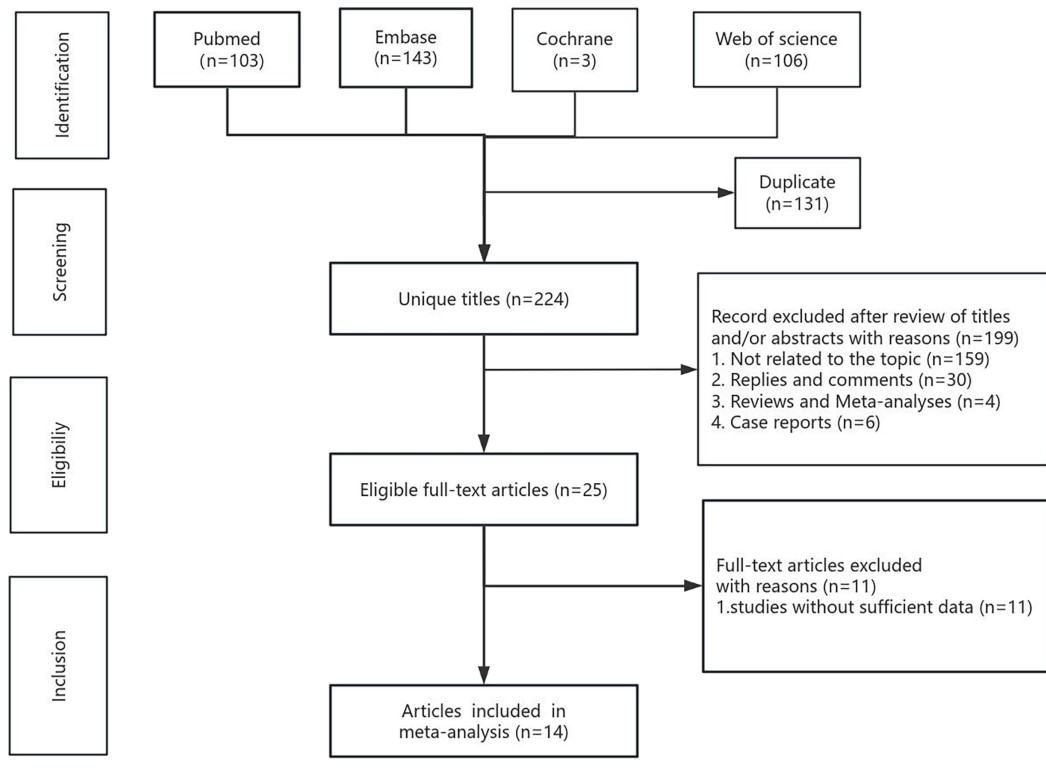

**Figure 1  Flowchart of the systematic search and selection process.**

[0.66–0.99], $P = 0.04$) and other subtypes (HR = 0.40, 95% CI [0.22–0.72], $I^2 = 89\%$, $P < 0.00001$), but not to pancreatic head cancer (HR = 0.46, 95% CI [0.16–1.13], $I^2 = 59\%$, $P = 0.12$). LMR retained its predictive value across different regions, including Asia (HR = 0.62, 95% CI [0.47–0.76], $I^2 = 68\%$, $P < 0.0001$), Europe (HR = 0.78, 95% CI [0.67–0.91], $I^2 = 0\%$, $P = 0.002$), and the Americas (HR = 0.14, 95% CI [0.08–0.24], $I^2 = 0\%$, $P < 0.00001$). Notably, both LMR cut-off values greater than or equal to three (HR = 0.62, 95% CI [0.47–0.82], $I^2 = 67\%$, $P = 0.0009$) and less than three (HR = 0.47, 95% CI [0.32–0.69], $I^2 = 85\%$, $P = 0.0001$) exhibited prognostic significance. After categorizing the 14 studies based on NOS scores as ≥7 (HR = 0.66, 95%CI [0.54–0.80], $I^2 = 53\%$, $P = 0.04$) and <7 (HR = 0.41, CI [0.23–0.72], $I^2 = 89\%$, $P < 0.00001$), the results showed that LMR had a good predictive value in both subgroups. The sensitivity analysis for OS confirmed the strong predictive value of LMR, whereas LMR did not exhibit predictive significance for RFS (HR = 0.35, 95% CI [0.09–1.32], $I^2 = 95\%$, $P = 0.12$). The subgroup analysis results of LMR for resectable pancreatic cancer are shown in Table 2.

## Overall survival

Overall Survival (OS) data was synthesized from a total of 14 studies, encompassing 4,019 patients. The pooled analysis revealed a statistically significant result, indicating that LMR plays predictive role in individuals diagnosed with resectable pancreatic cancer. When a patient's LMR surpassed the designated threshold, it was associated with a more positive prognosis, leading to an extended OS. Conversely, when a patient's LMR value fell below

**Table 2 Subgroup analysis of LMR for resectable pancreatic cancer.**

| Subgroup | LMR | | | | |
|---|---|---|---|---|---|
| | Study | No. of patients | OR[95%CI] | P value | I² |
| **Total** | 14 | 4,019 | 0.55[0.44–0.69] | <0.00001 | 79% |
| **Population** | | | | | |
| Pancreatic ductal adenocarcinoma | 5 | 1,556 | 0.73[0.57–0.93] | 0.01 | 46% |
| Pancreatic head cancer | 2 | 197 | 0.42[0.16–1.13] | 0.12 | 59% |
| Pancreatic Neuroendocrine Neoplasms | 1 | 174 | 0.81[0.66–0.99] | 0.04 | NA |
| Others | 6 | 2,092 | 0.40[0.22–0.72] | 0.002 | 89% |
| **Region** | | | | | |
| Asia | 8 | 1.312 | 0.60[0.47–0.76] | <0.0001 | 68% |
| Europe | 4 | 2,403 | 0.78[0.67–0.91] | 0.002 | 0% |
| America | 2 | 304 | 0.14[0.08–0.24] | <0.00001 | 0% |
| **LMR threshold** | | | | | |
| <3 | 8 | 3,108 | 0.47[0.32–0.69] | <0.00001 | 85% |
| ≥3 | 6 | 911 | 0.62[0.47–0.82] | 0.0009 | 67% |
| **NOS** | | | | | |
| <7 | 6 | 2,186 | 0.41[0.23–0.72] | <0.00001 | I² = 89% |
| ≥7 | 8 | 1,833 | 0.66[0.54–0.80] | 0.04 | I² = 53% |

**Note:**
   LMR, lymphocyte to monocyte ratio; OR, odds ratio; CI, confidence interval.

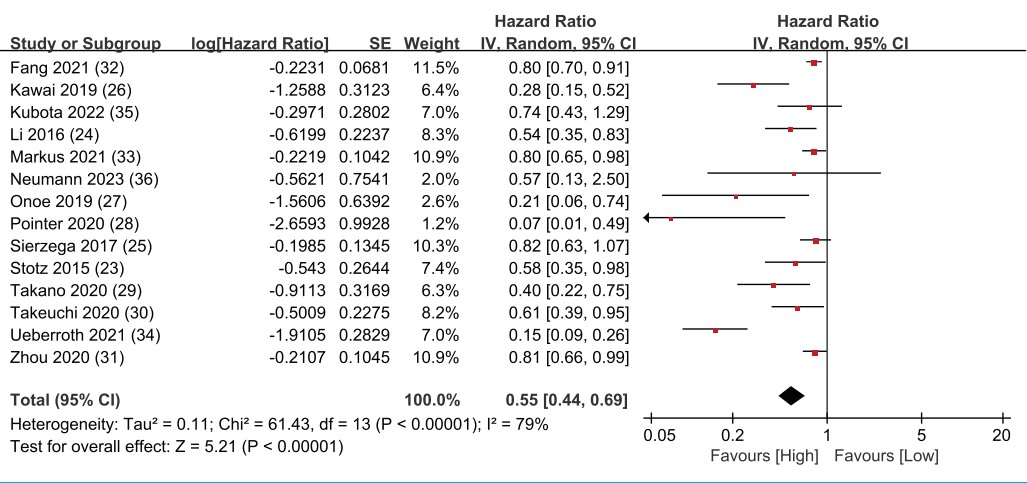

**Figure 2 Forest plots of OS outcomes.**

the LMR threshold, the patient's prognosis tended to be less favorable, leading to a shorter OS (HR = 0.55, 95% CI [0.44–0.69], I² = 79%, $P$ < 0.00001; Fig. 2).

## Subgroup analysis of OS: population

To further assess the prognostic significance of LMR for overall survival in various subtypes of pancreatic cancer, the 14 articles were divided into four different subtypes of pancreatic cancer for detailed investigation: pancreatic ductal adenocarcinoma, pancreatic

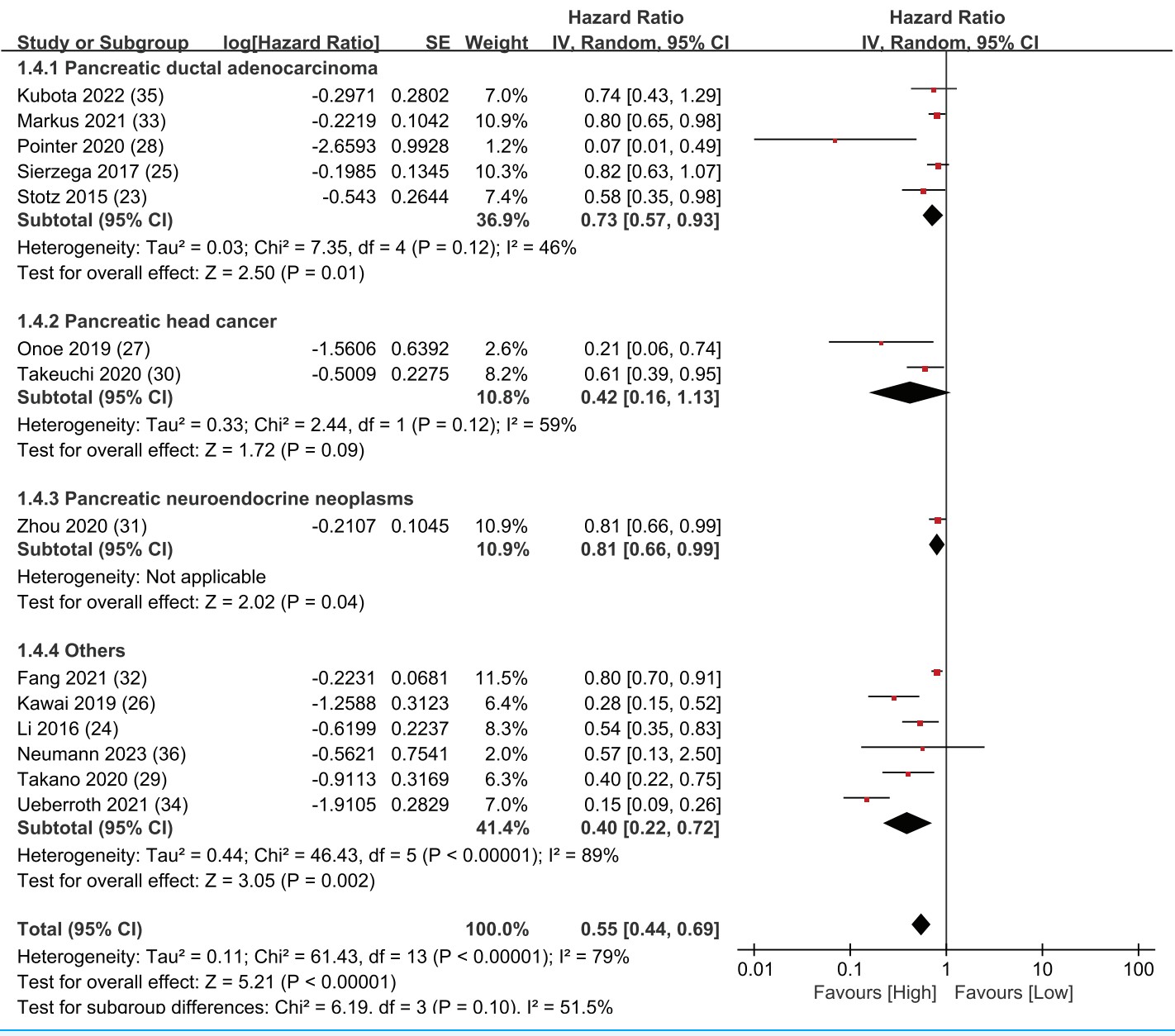

**Figure 3  Forest plots of subgroup analysis of OS: population.**  

neuroendocrine neoplasms, pancreatic head cancer, and all other types of pancreatic cancer (Fig. 3). In the pancreatic ductal adenocarcinoma group, five studies were included with statistically significant results (HR = 0.73, 95% CI [0.57–0.93], $I^2$ = 46%, $P$ = 0.01). There were two studies in the pancreatic head cancer group, and the results were not statistically significant (HR = 0.46, 95% CI [0.16–1.13], $I^2$ = 59%, $P$ = 0.12). The pancreatic neuroendocrine neoplasms group only included one study, but the results were statistically significant (HR = 0.81, 95% CI [0.66–0.99], $P$ = 0.04). Finally, the group of all other types of pancreatic cancer included six studies with statistically significant results (HR = 0.40, 95% CI [0.22–0.72], $I^2$ = 89%, $P$ < 0.00001). The Egger's test indicated publication bias within

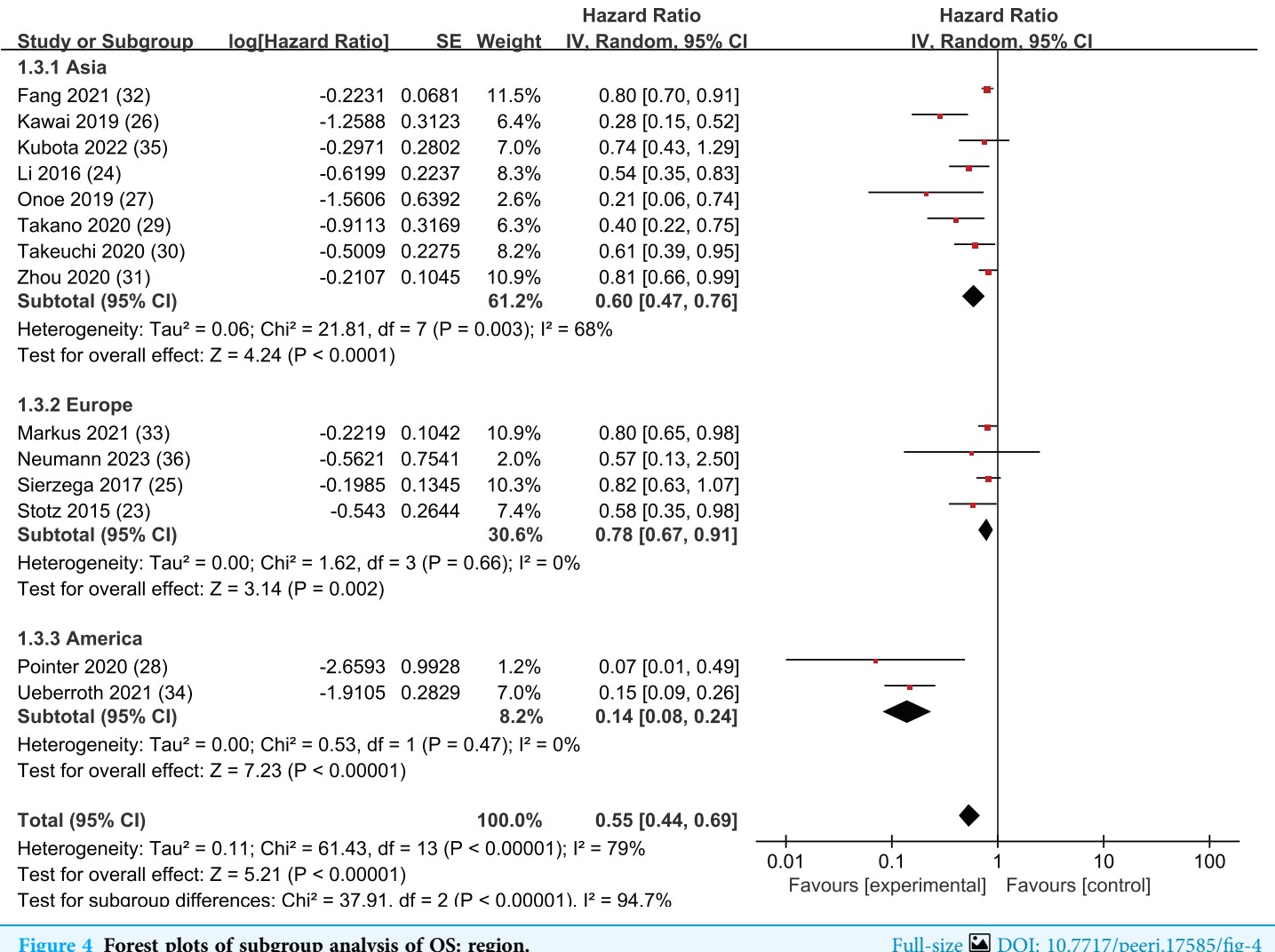

Figure 4 Forest plots of subgroup analysis of OS: region.               

the group of pancreatic ductal adenocarcinoma ($P = 0.041$) but not in the group of other types of pancreatic cancer ($P = 0.076$). The sub-group analysis findings indicate that LMR exhibits a robust predictive capacity for OS in pancreatic ductal adenocarcinoma, pancreatic neuroendocrine neoplasms, and other types of pancreatic cancer. However, the results for pancreatic head cancer were not statistically significant.

## Subgroup analysis of OS: region

A regional subgroup analysis was also performed based on the OS data from the 14 articles (Fig. 4). The studies were categorized into three major regions: Asia, the Americas, and Europe. There were eight articles from Asia, four articles from Europe, and two articles from the Americas. LMR showed prognostic significance in all three regional subgroup analyses: Asia (HR = 0.62, 95% CI [0.47–0.76], $I^2$ = 68%, $P < 0.0001$), the Americas (HR = 0.78, 95% CI [0.67–0.91], $I^2$ = 0%, $P = 0.002$), and Europe (HR = 0.78, 95% CI [0.67–0.91], $I^2$ = 0%, $P = 0.002$). The Egger's test revealed publication bias in the Asia

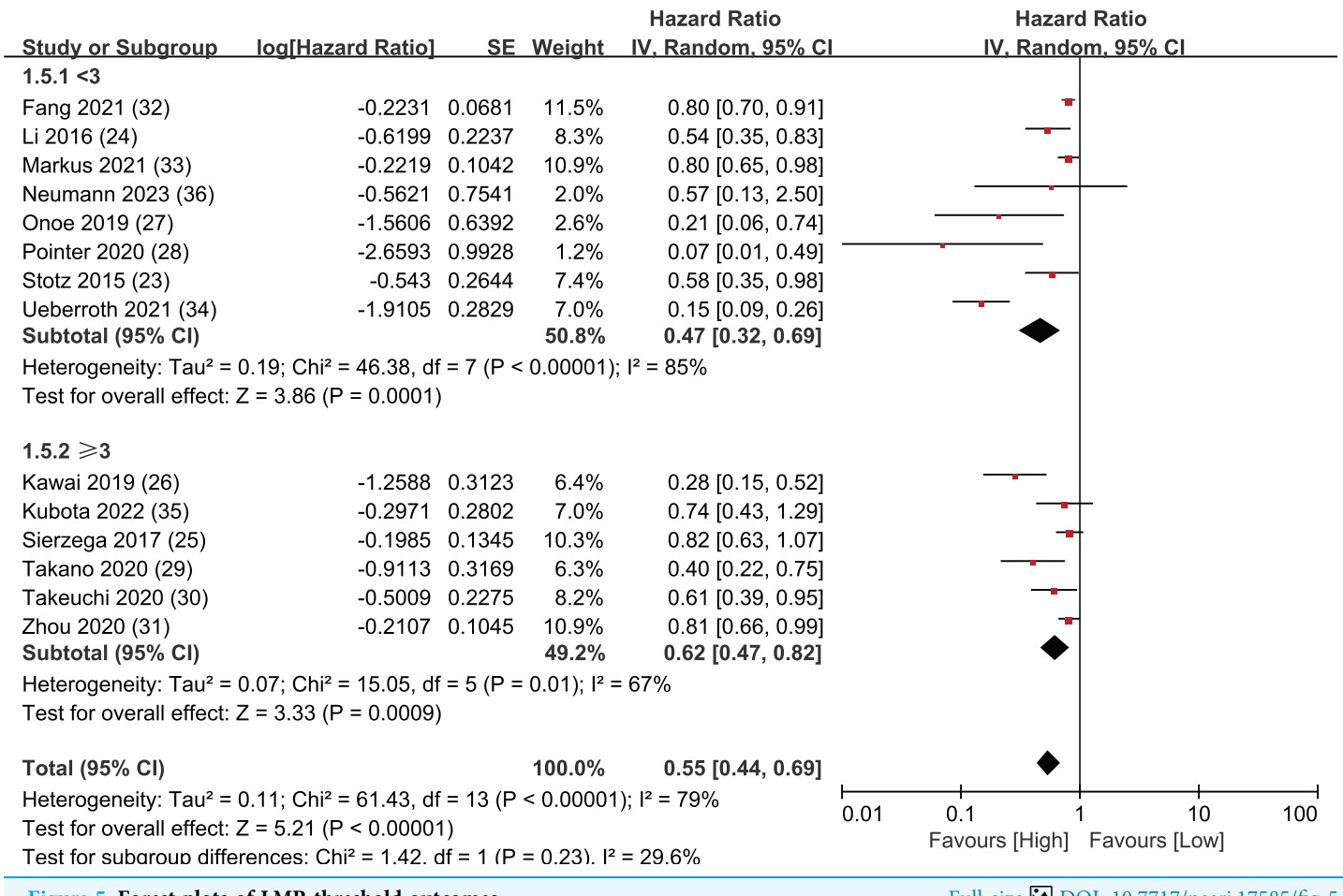

**Figure 5  Forest plots of LMR threshold outcomes.**               

subgroup ($P = 0.005$) but not in the Europe subgroup ($P = 0.242$). This indicates that LMR plays a significant predictive role in pancreatic cancer patients across these three continents.

## LMR threshold

LMR values were then extracted from the 14 articles and a threshold was set at three. This threshold value was chosen because eight of the 14 articles had LMR values greater than or equal to three, while six articles had LMR values less than three, making the groups close to equal. Separate analyses were performed for the eight articles and the six articles.

The results of these analyses showed that both the group with LMR values greater than or equal to three (HR = 0.62, 95% CI [0.47–0.82], $I^2 = 67\%$, $P = 0.0009$) and the group with LMR values less than three (HR = 0.47, 95% CI [0.32–0.69], $I^2 = 85\%$, $P = 0.0001$) had significant statistical significance. The Egger's test results demonstrated publication bias both in the group with LMR values greater than or equal to 3 ($P = 0.042$) and in the group with LMR values less than 3 ($P = 0.036$).These findings imply that establishing an LMR threshold of three is a reliable way to effectively forecast the prognosis of patients with pancreatic cancer (Fig. 5).

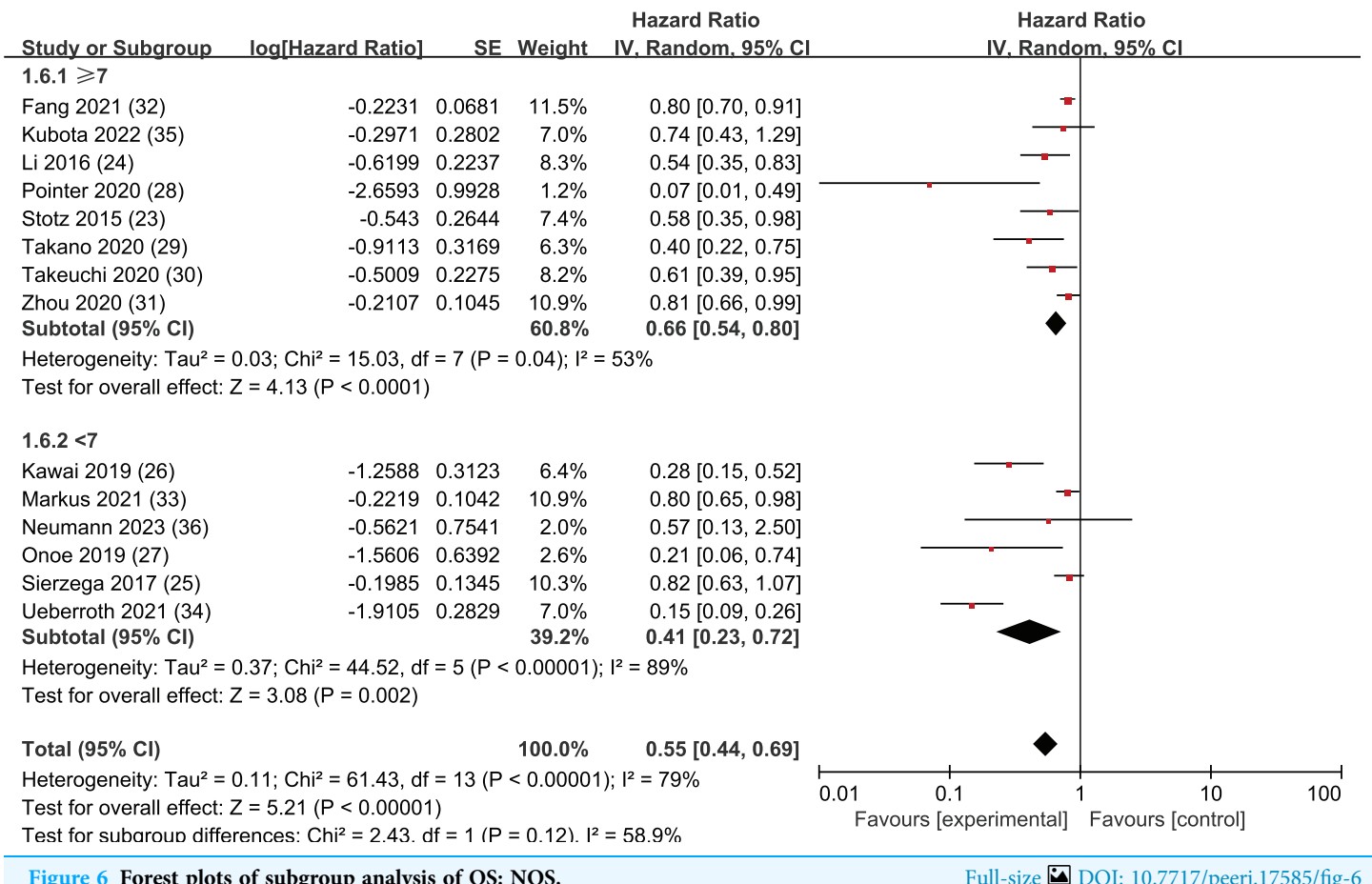

**Figure 6 Forest plots of subgroup analysis of OS: NOS.**

## NOS

In order to ensure a comprehensive analysis and maintain the integrity and representativeness of the data, we chose to include six studies of moderate quality. According to the Newcastle-Ottawa Scale (NOS) scores, five of these studies scored 6 points, while one study scored 5 points. To ensure the reliability of the moderate-quality studies, we categorized the articles based on Newcastle-Ottawa Scale (NOS) scores. Studies with NOS scores of 7 or higher were classified separately from those with scores below 7. Subgroup analyses were then conducted to compare the outcomes between these two groups.

The results revealed that among the 14 selected studies, both those with NOS scores greater than or equal to 7 (HR = 0.66, 95% CI [0.54–0.80], $I^2$ = 53%, $P$ = 0.04) and those with scores less than 7 (HR = 0.41, CI [0.23–0.72], $I^2$ = 89%, $P < 0.00001$) demonstrated statistically significant findings. The Egger's test results showed no publication bias in the group with NOS scores less than 7 ($P$ = 0.127), while publication bias was detected in the subgroup with NOS scores greater than or equal to 7 ($P$ = 0.002). These findings suggests that all 14 studies included in our analysis were reliable (Fig. 6).

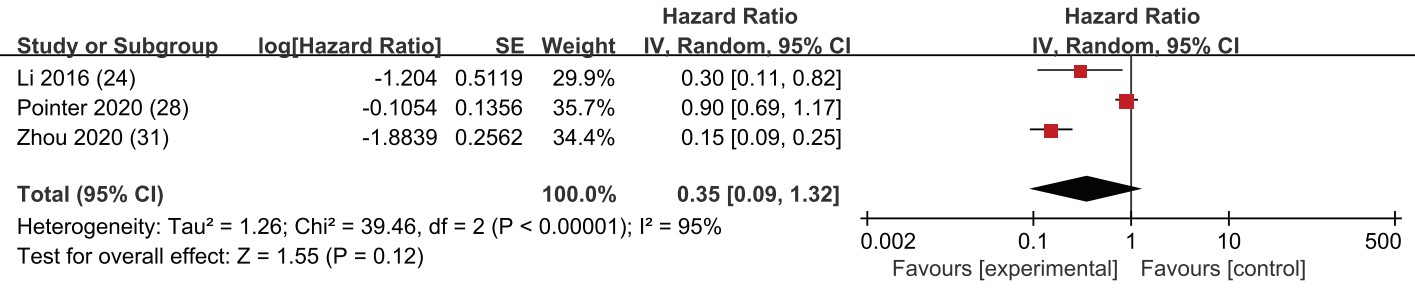

**Figure 7 Forest plots of RFS outcomes.**               

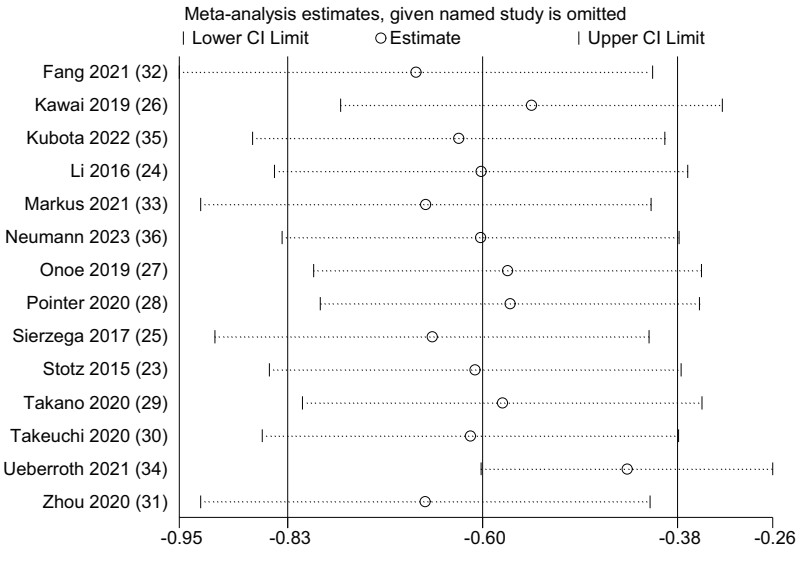

**Figure 8 Sensitivity analysis of OS.**     

## RFS

Among the 14 included articles, three provided HR along with corresponding confidence intervals for RFS. These data were extracted for analysis, and the results indicated that the *p*-values associated with RFS did not achieve statistical significance (HR = 0.35, 95% CI [0.09–1.32], $I^2$ = 95%, *P* = 0.12). This result suggests that LMR may not be a robust predictor of RFS in this context (Fig. 7). However, it is important to acknowledge that the absence of statistical significance in these studies may stem from limited datasets, and there could be some uncertainty in the statistical outcomes. Notably, an Egger's test did not raise significant concerns (*p* = 0.507).

## Sensitivity analysis

A univariate sensitivity analysis was performed for OS by systematically excluding individual studies to assess the impact of each study on the prognostic significance of LMR (Fig. 8). The outcomes of the sensitivity analysis indicated that, regardless of the exclusion of any specific study, the statistical significance of LMR in predicting OS remained stable

and unaltered. However, the results also indicated that (*Ueberroth et al., 2021*) may have some bias towards the upper confidence interval. This study tried to compare the high and low levels of LMR and NLR in Black and non-Black patients, which may have affected the confidence interval. However, the results of an Egger's test showed a high degree of publication bias ($p = 0.003$).

## DISCUSSION

This comprehensive meta-analysis incorporated 14 scholarly articles involving a total of 4,019 individuals afflicted by pancreatic cancer. The analysis revealed a substantial and noteworthy predictive capacity of peripheral blood levels of LMR in predicting the prognosis of patients with resectable pancreatic cancer.

Since Virchow initially proposed a close relationship between inflammation and cancer in 1863, accumulating evidence has supported this concept (*Zhou et al., 2020*). Emerging studies have additionally underscored that inflammation associated with tumors represents the seventh hallmark of cancer, exerting its influence throughout all phases of tumor initiation and progression (*Hanahan & Weinberg, 2011*; *Raucci et al., 2019*). Inflammatory biomarkers can serve as indicators of systemic inflammation severity (*Stone & Beatty, 2019*). *Mei et al. (2017)* provided evidence demonstrating a strong association between elevated neutrophil-to-lymphocyte ratio (NLR) and LMR in patients with resectable pancreatic cancer and a decrease in median survival. Research conducted by Bingle et al. demonstrated a consistent association between elevated macrophage density and unfavorable prognostic outcomes across a spectrum of malignancies, including breast cancer, prostate cancer, lung cancer, cervical cancer, and bladder cancer (*Bingle, Brown & Lewis, 2002*). In their study, *Condeelis & Pollard (2006)* noted that macrophages serve a multifaceted role in promoting tumor progression within the realm of medical research. This includes aiding in tumor cell invasion, promoting tumor cell migration and intravasation, facilitating tumor angiogenesis, and even exerting inhibitory effects on immune responses targeting tumor growth (*Pollard, 2004*; *Condeelis & Pollard, 2006*). Colony-stimulating factor 1 (CSF-1) plays a crucial role as a central regulator within the macrophage lineage. Changes in the levels of this factor have been linked to adverse prognoses in various cancer types, including colorectal cancer (*Mroczko et al., 2007*).

*Porrata et al. (2012a*, *2012b)* identified LMR as a prognostic marker in classical Hodgkin lymphoma and nodular lymphocyte-predominant Hodgkin lymphoma. This meta-analysis has likewise substantiated the favorable prognostic significance of LMR in pancreatic cancer. However, it is important to note there is no universally consistent threshold value for LMR across all studies to date.

This meta-analysis reintegrated six new pancreatic cancer articles from the past 6 years, distinguishing this analysis from that of *Li et al. (2017)* and expanding the patient cohort of the *Li et al. (2017)* analysis by 2,224 individuals. This substantial increase has significantly enhanced the accuracy of the predictive model. Additionally, subgroup analyses were performed on different pancreatic cancer types, yielding noteworthy predictive outcomes. Diverging from the study conducted by *Lin et al. (2020)*, this analysis distinguishes itself by incorporating three highly representative articles from the past 4 years into the analysis.

It also includes regional subgroup analyses of Asia, Europe, and the Americas, thus enriching the meta-analyses in this field with a more comprehensive perspective.

This analysis offers the latest understanding of the role of LMR in forecasting the prognosis of patients with resectable pancreatic cancer. However, this analysis also has certain limitations.

Firstly, all the studies included in this analysis were retrospective, as prospective studies were lacking after excluding non-compliant articles. This limitation arises from the homogeneity of study types. Secondly, when extracting data for RFS, only three articles provided HR for inclusion. Consequently, the conclusions regarding RFS may be less robust compared to those for OS. Thirdly, there was considerable heterogeneity in OS, and sensitivity analyses to evaluate its robustness revealed that the origins of heterogeneity for specific outcomes from *Ueberroth et al. (2021)* may have affected the upper limit of the confidence interval. *Ueberroth et al. (2021)* divided patients into Black and non-Black patients, and 163 patients were tissue-diagnosed with pancreatic adenocarcinoma and included in the analysis. Twenty-seven patients identified themselves as "Black"; 136 were analyzed as "non-Black," with the majority being considered "white." Black patients exhibited a "favorable" white blood cell profile (high LMR, low NLR) compared to non-Black patients, which may have affected the confidence interval. Given the potential influence of confounding factors, the findings of this meta-analysis should be approached with prudence when drawing conclusions. Fourthly, in the population sub-analysis, there was only one article available for pancreatic neuroendocrine neoplasms, which limited the ability to establish reference values, and additional data collection and consolidation is needed in this regard. Lastly, this analysis was only able to obtain the region of the original included studies but not the specific ethnicity of each patient included.

Despite the limitations of this analysis, it is important to highlight its significant strengths. Firstly, this study is the most current meta-analysis on the topic and boasts the largest sample size among all meta-analyses conducted to date. The inclusion of nine additional, previously unexamined articles spanning from 2020 to 2023 has substantially bolstered the credibility of the evidence base, setting it apart from earlier investigations. Secondly, a comprehensive analysis was performed, including a sensitivity analysis, subgroup analyses, and the application of Egger's test to assess HR related to OS. This comprehensive analysis reaffirms the clinical relevance of LMR in offering valuable prognostic insights for individuals who have been diagnosed with resectable pancreatic cancer.

## CONCLUSION

In the context of resectable pancreatic cancer, multiple studies consistently provide compelling evidence of the significant predictive utility of LMR in assessing patient prognosis. This conclusion aligns with earlier published studies and contributes to the body of evidence in evidence-based medicine, further substantiating the utility of LMR as a tool for the early identification of high-risk patients with pancreatic cancer.

### Funding

This work was supported by the Programs from the Department of Science and Technology of Anhui Province (Grant Number 2023AH051991) and the National Innovation and Entrepreneurship Project for college students (Grant Number 202210367037). The funders had no role in study design, data collection and analysis, decision to publish, or preparation of the manuscript.

### Grant Disclosures

The following grant information was disclosed by the authors:
Department of Science and Technology of Anhui Province: 2023AH051991.
National Innovation and Entrepreneurship Project: 202210367037.

### Competing Interests

The authors declare that they have no competing interests.

### Author Contributions

- Haipeng Li conceived and designed the experiments, performed the experiments, analyzed the data, prepared figures and/or tables, authored or reviewed drafts of the article, and approved the final draft.
- Shang Peng performed the experiments, authored or reviewed drafts of the article, and approved the final draft.
- Ran An performed the experiments, authored or reviewed drafts of the article, and approved the final draft.
- Nana Du analyzed the data, prepared figures and/or tables, and approved the final draft.
- Huan Wu analyzed the data, prepared figures and/or tables, and approved the final draft.
- Xiangcheng Zhen conceived and designed the experiments, prepared figures and/or tables, and approved the final draft.
- Yuanzhi Gao performed the experiments, prepared figures and/or tables, and approved the final draft.
- Zhenghong Li conceived and designed the experiments, authored or reviewed drafts of the article, and approved the final draft.
- Jingting Min conceived and designed the experiments, authored or reviewed drafts of the article, and approved the final draft.

### Data Availability

This is a systematic review/meta-analysis.

### Supplemental Information

Supplemental information for this article can be found online at http://dx.doi.org/10.7717/peerj.17585#supplemental-information.

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
