# Peer review of "The prognostic role of lymphocyte-to-monocyte ratio in patients with resectable pancreatic cancer: a systematic review and meta-analysis"

_PeerJ, doi:10.7717/peerj.17585_

## Round 0.1 · original submission · Major Revisions

Dear authors

On the basis of two reviewers who recommended that the work requires extensive corrections, a decision was made that it is necessary to change your work in detail and correct it according to the opinion of the reviewers.

**Language Note:** The review process has identified that the English language must be improved. PeerJ can provide language editing services - please contact us at [email protected] for pricing (be sure to provide your manuscript number and title). Alternatively, you should make your own arrangements to improve the language quality and provide details in your response letter. – PeerJ Staff

Reviewer 1 ·

Basic reporting

The study by Li et al, focused on assessing the predictive potential of the Lymphocyte-to-Monocyte Ratio (LMR) in preoperative pancreatic cancer patients, demonstrates a comprehensive and systematic approach to understanding this prognostic marker. By conducting an extensive literature search across reputable databases, including PubMed, Embase, Cochrane, and Web of Science, the authors aimed to provide a robust evaluation of LMR's prognostic significance. This study could be a useful addition to the literature on LMR in preoperative pancreatic cancer and advances current meta-analysis already found in the field as cited and discussed by the authors. The introduction and background provided are relevant to the research study.

However, there are some concerns that I would like to raise:

The article needs some major rewriting, and could benefit from proofreading by a person who is proficient in English or a professional editing service. Past tense should be followed throughout the whole article and not present tense (for example ll 82, 91). Punctuation mistakes can be found throughout the article and should be corrected. For the inconsistency index I2 please ensure that the 2 is in superscript throughout the manuscript ( for example ll 140, 168, etc).
Please keep consistency in the way you are writing lymphocyte-to-monocyte ratio (with or without “-“). Also, the authors sometimes jump from LMR to MLR which makes it confusing to the reader. Consistency in this regard would be beneficial to bring the research outcomes across.
The conclusion sentence in the general overview should be reworded to “Despite of the observed heterogeneity and biases..” and not “In light of..”.

The whole paragraph for “subgroup analysis of OS: population” needs to be rewritten. Sentences need to be reworded to adhere to the professional English language. Words such as “grim”, “..is excellent..”, “..is good..” or “..is not so good..” or “…not so bad..” should not be found in a scientific article. Please rewrite these sections to not attach any value to the findings but to keep the article in a neutral and professional tone. What do the authors mean when they say a result “is excellent” or “not so good”?

Ll 181 the authors talk about 14 different subtypes – which subtypes are they referring to?

As per PeerJ guidelines, the authors should provide the registry name and number/URL of their study in the abstract.

Experimental design

The authors give a clear overview of the search functions that were used to find relevant articles for this meta-analysis and the criteria they were using to find for eligible studies to be included in this meta-analysis.
The flow chart they provide is helpful. However, the numbers in the flow chart do not match the numbers matched in the free text. This needs to be updated. Additionally, there typos in the flow chart (and other figure elements i.e. figure 2). Please correct these. Could the authors please explain what “on clamp” and “off clamp” techniques are?

The authors claim that for the pancreatic head cancer only 2 studies were included and that therefore the results may not be reliable. However, for the neuroendocrine subtype only one study was included and here the authors do not raise any concerns. A better metric to how many studies were included would be to show how many patients were part of each study. It would be beneficial to include a table that shows how many patients were part of each of the studies that were included in this meta-analysis.
The authors are contradicting themselves: in ll 178/179 they state there was only 1 study available for neuroendocrine pancreatic cancer but in ll186 they are mentioning 2. Please correct this.

The authors state the LMR threshold was set to 3 (ll 196 following) but give no reasoning as to why this particular threshold was chosen which makes it seem that this threshold was chosen randomly. A more detailed explanation in this regard would be useful.

Given the current emphasis on diversity and inclusion in medical research I think the authors did a good job in including OS analysis based on geographic regions. However, I would like to raise the concern that just because a study was conducted in Europe, this does not automatically mean that every patient included in the study was of European descent. Would it be possible for the authors to calculate OS based on the ancestry of the patients based in the studies rather than on geographical location of where the study took place? Alternatively, if this is not possible, could the authors please include a statement addressing the limitation as outlined above?

Validity of the findings

The study suggests that LMR may serve as a valuable predictor of OS in resectable pancreatic cancer patients. The thorough methodology, consideration of diverse factors, and acknowledgment of limitations contribute to the overall findings of the study. I commend the authors for clearly stating some limitations of their study.

“A visual examination of the funnel plot revealed a slight indication of publication bias
(Figure 2). Nevertheless, it is noteworthy that the results of Egger’s test did not achieve statistical significance (p = 0.003).” The authors are stating they are seeing a slight skewing of the funnel plot pointing towards publication bias. When running an Egger’s test, the p-value of this test is highly significant, which means that the authors are dealing with a significant publication bias in their study. Please correct your statement to represent your results. Also, the authors are stating that Figure 2 represents a funnel plot, however the displayed graph is a forest plot. Please provide a funnel plot for your analysis.

“The outcomes of the sensitivity analysis indicate that, regardless of the
exclusion of any specific study, the statistical significance of LMR in predicting overall survival (OS) remained stable and unaltered.” While this statement seems to be true for 13 out of the 14 articles, it appears that for Ueberroth 2021 there may be some skewing towards the Upper CI limit. Could the authors please comment on this and how this would affect the lower CI if this study was omitted?

Reviewer 2 ·

Basic reporting

The manuscript demonstrates a generally good English language, although some revisions are needed for accuracy and professional expression (See additional comments). The flow of the manuscript is smooth, and the literature is appropriately referenced.

Experimental design

The research is well-designed, employing detailed and appropriate methods to address the research question, despite the existence of similar publications.
Minor comments:
1. Additional details are required regarding the inclusion criteria, particularly regarding the pathological diagnosis of pancreatic cancer for included patients, the definition of "eligible for surgical removal" (e.g., resectable, stage I-III, or receive surgical treatment), and clarification on whether HR is for survival outcomes (OS and RFS).
2. The inclusion of a meta-regression analysis would be beneficial to assess whether subgroup factors contribute to the observed heterogeneity.

Validity of the findings

The article employs solid methods and effectively addresses the research question of interest.
Minor comments:
1. A table outlining the features of the included studies should be provided. It should include basic information such as primary author, publication year, study location, sample size, participant demographics (age/sex), follow-up duration, treatment modalities, LMR cut-off, and outcomes evaluated (OS or PFS, multivariate or univariate analysis), types of pancreatic cancer and NOS score.
2. In addition to reporting I2 and p-values, HR and corresponding 95% CI should be presented in both the results and abstract.
3. The results of the subgroup analysis could be synthesized into a table for improved conciseness, with inclusion of the number of patients in each subgroup analysis.
4. A general summary of LMR cut-off ranges, NOS scores, and evaluated outcomes should be included in the results section under study characteristics.

Additional comments

Sentences requiring revision or check
(Abstract)
Line 15: Consider revising "preoperative pancreatic cancer patients" to "patients with resectable pancreatic cancer" or utilize "preoperative/pre-treatment LMR" for improved professional expression.
Lines 28-29: The accuracy of these sentences needs improvement. For example, revise it as "Notably, both LMR cut-off values greater than or equal to 3 and less than 3 exhibit prognostic significance." Ensure that the terms "prognostic" (for outcome) and "predictive" (for treatment response) are appropriately used throughout the manuscript.

(Introduction)
Lines 41-42: This sentence appears to lack logical flow with the preceding text. Additionally, consider adding a brief review of existing pancreatic cancer biomarkers in this section.

(Method)
Lines 82-98: The specific search queries may not be necessary within the article body, given that the keywords were provided.

(Result)
Line 156: The number 342 here doesn’t match the number 224 in the figure 1.
Lines 169-171: The figure 2 is forest plot rather than funnel plot. The statement "not achieve significance" contradicts with "p= 0.003." Consider moving the result of publication bias to the place of the sensitivity analysis, to be organized.
Lines 202-203: Given the significant p-value observed in both LMR cut-off greater and less than 3 subgroups, it is not accurate to make the conclusion about reliability of an LMR threshold of 3.

(Discussion)
Line 223-224: This study is not the first meta-analysis about the prognostic value of LMR in pancreatic cancer.

(Figures)
Recommend citing literature references in all figures. In Figure 1, numbers of studies excluded for the specific reasons should be provided; figure 1 "Studies without comparison of on-clamp vs off-clamp techniques" need to be checked

Other sentences requiring revision for grammar or expression: Lines 18-22, 43, and 192-194.

---

## Round 0.2 · Minor Revisions

The prognostic role of lymphocyte-to-monocyte ratio in patients with resectable pancreatic cancer: a systematic review and meta-analysis - that it requires a few Minor Revisions.

Reviewer 1 ·

Basic reporting

The authors have addressed the majority of my concerns and have significantly improved the manuscript in the English language, making it much easier to follow their thought process and results.

Experimental design

Minor comments:

The authors say that a NOS score of 7-9 states a high-quality study.
However, in their results section they are mentioning that they included studies with a NOS score of 6 (l 426). Could the authors please leave a comment as to why lower quality studies were included?

The resolution of some of the figures is very low. Please ensure that this will be addressed prior to publication.

Figure 2: for pointer et al the authors drew an arrow instead of a simple line. Please correct.

Validity of the findings

N/a

Reviewer 3 ·

Basic reporting

passed

Experimental design

passed

Validity of the findings

passed

Additional comments

The authors addressed most comments from reviewers and the manuscript was much improved.

Some Extra points:
- Does each study included had the same adjusted variables (age, sex, race, etc.) for LMR (do we know)? As these covariates can much affect Cox regression coefficient of LMR.
- Reviewer 1, comment 10 – did not see the funnel plot corresponding to Egger’s test of p=0.003. Also, did you run this test for the whole dataset, and other subgroup analysis (and what are the results)?
- For subgroup analysis, what about sex or other variables (which were not yet in table 2)?
- Figure image quality can be improved (generally is blurred)

---

## Round 0.3 · accepted · Accept

The article is now Acceptable - congratulations